# Whom do we prefer to learn from in observational reinforcement learning?

Gota Morishita[1]*, Carsten Murawski[1], Nitin Yadav[1], Shinsuke Suzuki[1,2,3]*

**1** Centre for Brain, Mind and Markets, The University of Melbourne, Parkville, Victoria, Australia,
**2** Faculty of Social Data Science, Hitotsubashi University, Kunitachi, Tokyo, Japan, **3** HIAS Brain
Research Center, Hitotsubashi University, Kunitachi, Tokyo, Japan

\* gota.morishita@gmail.com(GM); shinsuke.szk@gmail.com(SS)

pcbi.1013143

**Editor:** Bastien Blain,
Pantheon-Sorbonne University: Universite
Paris 1 Pantheon-Sorbonne, FRANCE

**Peer Review History:** PLOS recognizes
the benefits of transparency in the peer
review process; therefore, we enable the
publication of all of the content of peer
review and author responses alongside
final, published articles. The editorial
history of this article is available here:
https://doi.org/10.1371/journal.pcbi.
1013143

## Abstract

Learning by observing others' experiences is a hallmark of human intelligence. While
the neurocomputational mechanisms underlying observational learning are well
understood, less is known about whom people prefer to learn from in the context of
observational learning. One hypothesis posits that learners prefer individuals who
exhibit a high degree of decision noise, 'free riding' on the costly exploration of others. An alternative hypothesis is that learners prefer individuals with low decision
noise, as lower decision noise is often associated with better performance. In a pre-
registered experiment, we found that most participants preferred to learn from low-
noise (high-performing) individuals. Furthermore, exploratory analyses revealed that
participants who preferred low-noise individuals tended to rely on imitation of others'
actions. These findings offer a potential computational account of how learning styles
are related to partner selection in social learning.

## Author summary

In our daily lives, we often learn by watching others. For example, when starting
a new job, we might watch an experienced colleague to learn effective strategies,
or an inexperienced coworker to avoid common mistakes. While previous studies have examined how people learn by observing others, less is known about
how we decide whom to observe. This study explored whether people prefer to
learn from individuals who make consistent choices or those who behave more
randomly. At first glance, the answer seems obvious: we would naturally prefer
to learn from those who make consistent, reliable decisions. However, there can
also be value in learning from someone who behaves unpredictably. For example, when searching for a good restaurant, observing an adventurous friend who
tries unfamiliar places might help us discover hidden gems. Despite this potential
advantage, we found that most participants preferred consistent decision-makers.

**Data availability statement:** All data and statistical analysis code are available on Zenodo at https://doi.org/10.5281/zenodo.15386571.

**Funding:** The author(s) received no specific funding for this work.

**Competing interests:** The authors have declared that no competing interests exist.

Further analysis revealed that participants who favored reliable partners tended to imitate their actions. Our findings suggest that personal learning styles shape partner preferences. These insights could help us understand how people choose whom to learn from in everyday settings like classrooms, or workplaces.

## Introduction

Observational learning, the process through which individuals acquire knowledge and skills by observing others, is a critical ability for social animals, including humans, to survive in uncertain environments [1,2]. For instance, Adelie penguins utilize observational learning to determine whether there are any predators in the sea before entering the water themselves to forage for food [3]. Instead of diving in right away, they watch how the first penguin reacts after diving in the water. If no predator appears, the rest follow, using this observation to judge whether the environment seems relatively safe. Similarly, humans often rely on observational learning in decision-making processes. For example, when choosing a restaurant, people often read online reviews or consider friends' recommendations using others' experiences to identify high-quality restaurants while minimizing the risk of choosing a poor dining experience [4].

Given the ubiquity of observational learning, numerous studies across disciplines have investigated its computational and neural mechanisms [5–16]. Research has shown that individuals use both reward outcomes and actions of others to guide their own decisions [8,9,11]. Specifically, when making choices, people not only rely on action values learned through observing the reward outcomes associated with others' actions (learning from reward outcomes of others), but they also exhibit a separate bias toward actions frequently chosen by others (learning from actions of others). However, most of these studies predetermined the partners that participants observed, offering limited insight into how learners choose whom to learn from—a crucial aspect in real-world settings, where individuals often have the autonomy to decide whom to learn from. For instance, in workplace settings, individuals may selectively observe experienced colleagues to identify actions that lead to good results, or less experienced peers to recognize and avoid ones that lead to bad results. Nonetheless, little is known about individuals' preferences for selecting observational partners.

To fill this gap, we examined individuals' preferences for observational partners, specifically focusing on their preferences regarding a partner's level of decision noise (i.e., randomness in action selection). From a computational perspective, decision noise modulates the degree of random exploration [17]: higher decision noise leads to greater exploration, whereas lower decision noise corresponds to reduced exploration. In sequential value-based decision-making, individuals face a fundamental trade-off between exploring unfamiliar options to gain information and exploiting familiar options to maximize immediate rewards [18]. Thus, observing partners with different levels of decision noise may influence how individuals learn and adjust their

own decision-making, as partners with higher or lower decision noise provide varying amounts of information about available options. Moreover, individual differences in learning styles may lead to differential selection of observational partners based on their decision-noise characteristics.

We hypothesized that individuals would prefer high-noise partners (i.e., partners who exhibit a high degree of exploration). Learning from such partners may provide advantages by allowing observers to gain information about a wider range of unfamiliar options without directly incurring risk. Economic theory supports this notion, demonstrating that observing highly explorative partners can yield greater informational benefits, ultimately leading to higher total rewards [19,20].

An alternative possibility is that individuals prefer low-noise partners. Such a preference would be advantageous if they imitate their partners' behavior, as low-noise partners tend to appear more consistent and successful, whereas high-noise partners may be perceived as less competent. Prior work on social learning shows that people are likely to imitate those viewed as more successful [21]. From this perspective, a competing hypothesis is that individuals would prefer low-noise (and thus high-performing) partners.

To distinguish between these competing hypotheses, we designed a novel behavioral experiment in which participants chose between two potential partners categorized as either high-noise or low-noise. Participants subsequently engaged in an observational learning task with their selected partner.

We found that the majority of participants exhibited a preference for selecting a low-noise (high-performing) partner. To examine individual differences in partner selection and learning styles, we conducted regression analyses along with model-based exploratory analyses. These analyses revealed that individuals who preferred low-noise partners demonstrated a greater reliance on imitative learning. These findings offer a computational perspective on the mechanisms underlying partner preference in social learning contexts, shedding light on how individuals determine whom to observe for informed decision-making.

## Results

### Overview of experimental design

We conducted two independent studies using the same experimental design: a pilot study and a main study. In both, the behavioral experiment comprised four blocks (see Methods for details). In each block, participants completed the Passive Observation task twice, and the Partner Selection and Observational Learning tasks once each (Fig 1). Prior to these four blocks, participants performed the Individual Learning task as a practice block, allowing them to become familiar with the basic task structure (see Methods for details).

In each of the two Passive Observation tasks, participants observed one potential partner performing a three-armed bandit task (i.e., repeatedly choosing one of the three options and receiving a reward based on the probability associated with the chosen option; Fig 1A) [22–24]. The potential partner differed between the first and second Passive Observation tasks: one demonstrated a high degree of decision noise, while the other exhibited a low degree of decision noise (Fig 1E). The presentation order of the two types of potential partners was counterbalanced across the four main blocks.

In the Partner Selection task, participants selected one of the two potential partners to learn from in the subsequent Observational Learning task (Fig 1B). During the Observational Learning task, participants performed a three-armed bandit task side by side with their chosen partner (Fig 1C). On each trial, participants observed their partner's action (i.e., selection of an option) and its outcome (i.e., rewarded or not) before making their own choice. Notably, the outcomes of the participants' own choices were not displayed, thereby preventing learning from direct reward experiences.

Importantly, the set of reward probabilities (0.25, 0.50, and 0.75) was identical in both the Passive Observation task and the subsequent Observational Learning task. However, the fractal stimuli and screen positions associated with these probabilities were randomized across tasks, both within and across blocks.

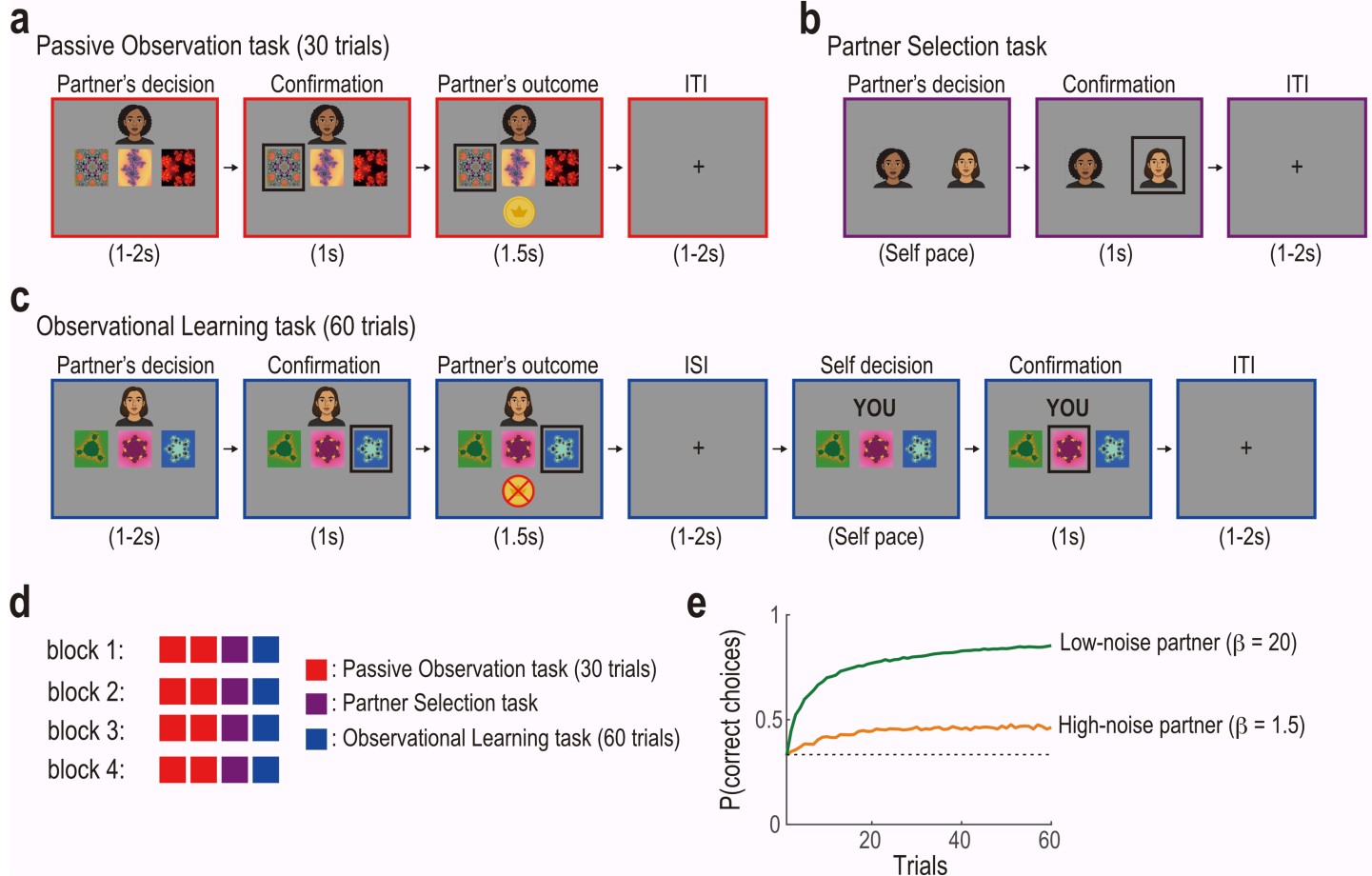

**Fig 1. Experimental task and basic behavior.** (a) Timeline of each trial in the Passive Observation task. Participants observed a potential partner choose between three options and receive a reward or not over 30 trials. (b) Partner Selection task. Participants chose between two potential partners they had observed in the Passive Observation task as the partner with whom they would perform the Observational Learning task. (c) Timeline of each trial in the Observational Learning task. Participants observed the partner's choice and outcome, and then made their own choice between the same set of options, without observing their own outcome over 60 trials. (d) Timeline of the entire experiment. In each block, participants completed two Passive Observation tasks, observed two potential partners with differing levels of decision noise, and then performed a Partner Selection task. Finally, they completed an Observational Learning task with the selected partner. The order of blocks was randomized across participants. (e) Proportions of correct choices (i.e., choosing the option with the highest reward probability) generated by a reinforcement learning algorithm with different degrees of decision-noise (i.e., inverse temperature $\beta$). The green line represents the low-noise partner's learning curve ($\beta = 20.0$), while the orange line represents the high-noise partner's curve ($\beta = 1.5$).

## Basic behavior

We first confirmed that participants learned to select the most rewarding option, verifying their comprehension of the task structure. The proportion of correct choices (i.e., choosing the option with the highest reward probability) significantly exceeded chance levels in the practice Individual and Observational Learning tasks, as well as in the main Observational Learning task (S1 Fig: average proportion over 60 trials = $0.73 \pm 0.03$ (Mean and SEM across participants), $t(55) = 12.99$, $p < 0.001$ for the practice Individual Learning task; proportion = $0.76 \pm 0.03$, $t(55) = 13.78$, $p < 0.001$ for the practice Observational Learning task; and proportion = $0.71 \pm 0.02$, $t(55) = 15.59$, $p < 0.001$ for the main Observational Learning task).

## Partner selection for observational learning

We next examined the main research question of whom individuals prefer to learn from in the context of observational learning. We found that the majority of participants preferred the low-noise partner over the high-noise partner (Fig 2). Pilot data ($N = 20$) showed that in the Partner Selection task, 13 out of 20 participants were more likely to select the low-noise partner (i.e., the proportion of selecting the high-noise partner across the four blocks was less than 0.5: see the green bars in Fig 2A), 5 participants selected the high-noise partner (i.e., the proportion was greater than 0.5: see the orange bars in Fig 2A), and 2 participants exhibited no preference (i.e., the proportion was equal to 0.5: see the white bar in Fig 2A).

To rigorously evaluate the findings of the pilot study, we conducted a preregistered main experiment. Consistent with the pilot data, participants in the main experiment demonstrated a preference for the low-noise partner in the Partner Selection task (Fig 2B and 2C). That is, most participants were more likely to select the low-noise partner to learn from

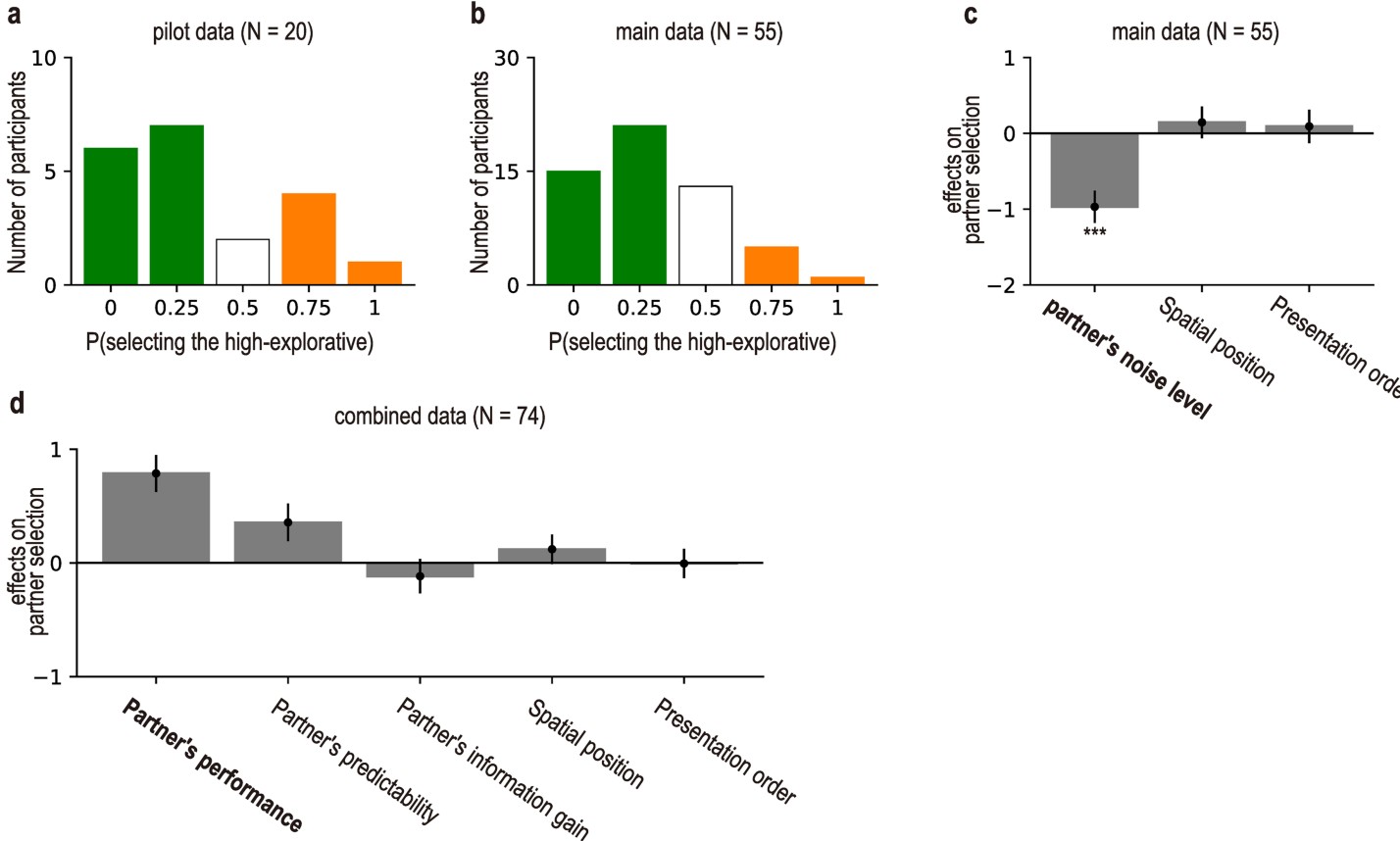

**Fig 2. Partner preference results.** (a) Histogram of the proportions of selecting the high-noise partner for the pilot data ($N = 20$). (b) Histogram of the proportions of selecting the high-noise partner for the main data ($N = 55$). In both (a) and (b), bar colors represent levels of preference: green bars indicate a low preference for selecting the high-noise partner, the white bar represents neutral preference (0.5), and orange bars indicate a high preference for selecting the high-noise partner. (c) Effects on partner selection in the main study (Mean ± SEM, $N = 55$). The leftmost bar shows a significant negative effect of the partner's noise level on selection probability ($p < 0.001$), indicating that participants were less likely to choose the high-noise partner. The other bars show non-significant effects of spatial position and presentation order. The means and SEMs were estimated with a generalized linear mixed model (GLMM). P-values were obtained using two-tailed t-tests. (d) Effects on partner selection in the combined data (Mean ± SEM, $N = 74$). We tested three factors underlying the decision noise manipulation: performance, predictability, and information gain of the partner. P-values are not reported as the analysis is exploratory.

for the subsequent Observational Learning task (proportion of selecting the high-noise partner over the four blocks was less than 0.5; Fig 2B). A mixed-effects logistic regression analysis revealed a significant negative effect of the partner's decision-noise level on partner selection ($b = -0.97 \pm 0.213, t(55) = -4.557, p < 0.001$; S1 Table), while controlling for a potential confounding effect of the spatial position of the partner (i.e., whether the potential partner was displayed on the left or right side of the screen in the Partner Selection task; see Fig 1B). This effect remained significant ($b = -1.06 \pm 0.25, t(55) = -4.27, p < 0.001$; Fig 2C), even after additionally controlling for another potentially confounding effect of the presentation order (i.e., whether the potential partner was presented in the first or second Passive Observation task). These results collectively show that most participants preferred to learn from the low-noise partner.

It is worth noting that decision noise can arise from multiple underlying factors, such as performance and predictability. To examine which factor best accounted for partner selection, we ran an exploratory mixed-effects logistic regression with three predictors: partner performance, predictability, and information gain (a proxy for directed exploration level) estimated from the Passive Observation tasks (see Methods for details). In this exploratory analysis, we combined data from the pilot and main studies, yielding $N = 74$ after excluding one pilot participant who completed only two of the four blocks. We reported effect sizes and standard errors rather than p-values, as p-values are more appropriate for statistical inference based on a priori hypotheses [25,26] (same for the subsequent analyses). The analysis showed that performance was the strongest positive predictor of partner selection ($b = 0.759 \pm 0.169$; Fig 2D), followed by a weaker effect of predictability ($b = 0.389 \pm 0.161$; Fig 2D). In contrast, information gain had no reliable effect ($b = -0.142 \pm 0.193$; Fig 2D). Taken together, these results suggest that participants preferentially selected low-noise partners primarily because those partners performed better.

## Individual differences in the partner selection and style of observational learning

The preregistered experiment above revealed that the majority of participants preferred learning from the low-noise (high-performing) partner, while others preferred the high-noise (low-performing) partner. However, what mechanism underlies the individual differences in partner selection for observational learning? We reasoned that these differences reflect participants' learning styles. Specifically, participants who preferred the low-noise partner may have relied more on learning from the partner's actions (i.e., imitation), as low-noise partners consistently made good choices, making the partners more consistent and reliable for imitation. In contrast, participants who preferred the high-noise partner may have relied more on learning from the partner's reward outcomes, since high-noise partners explored a broader range of options, enabling observers to learn about the rewards associated with different choices and discover better options themselves. To test this reasoning, we conducted exploratory regression analysis along with computational modeling analysis.

Using a generalized linear mixed model (GLMM), we assessed the extent to which each participant relied on learning from the partner's action and reward, as well as how this learning style was related to individual differences in partner selection. The GLMM quantified that, in the Observational Learning task, how much participants' trial-by-trial behavior was influenced by the partner's past action and reward. In addition to the main effects of the partner's past action and reward, the GLMM included the interaction terms with the individual difference in partner selection (i.e., the proportion of blocks in which the participant selected the high-noise partner in the Partner Selection task). Crucially, the interaction terms allowed us to quantify how learning styles (i.e., the effects of the partner's past action and reward) were associated with partner selection across participants.

The GLMM revealed that, on average, participants in the Observational Learning task learned from both the partner's past action and reward. The partner's past action and reward had positive effects on participants' behavior ($b = 0.83 \pm 0.06$ for the action effect; $b = 1.63 \pm 0.11$ for the reward effect; Fig 3A), consistent with prior research on observational learning [8,9,15]. Moreover, the effects of the partner's past action on participants' behavior in the Observational Learning task were modulated by individual differences in partner selection, as indicated by the interaction effect (Fig 3A and 3B). That is, the effect of the partner's past action was negatively modulated by the participants' proportion

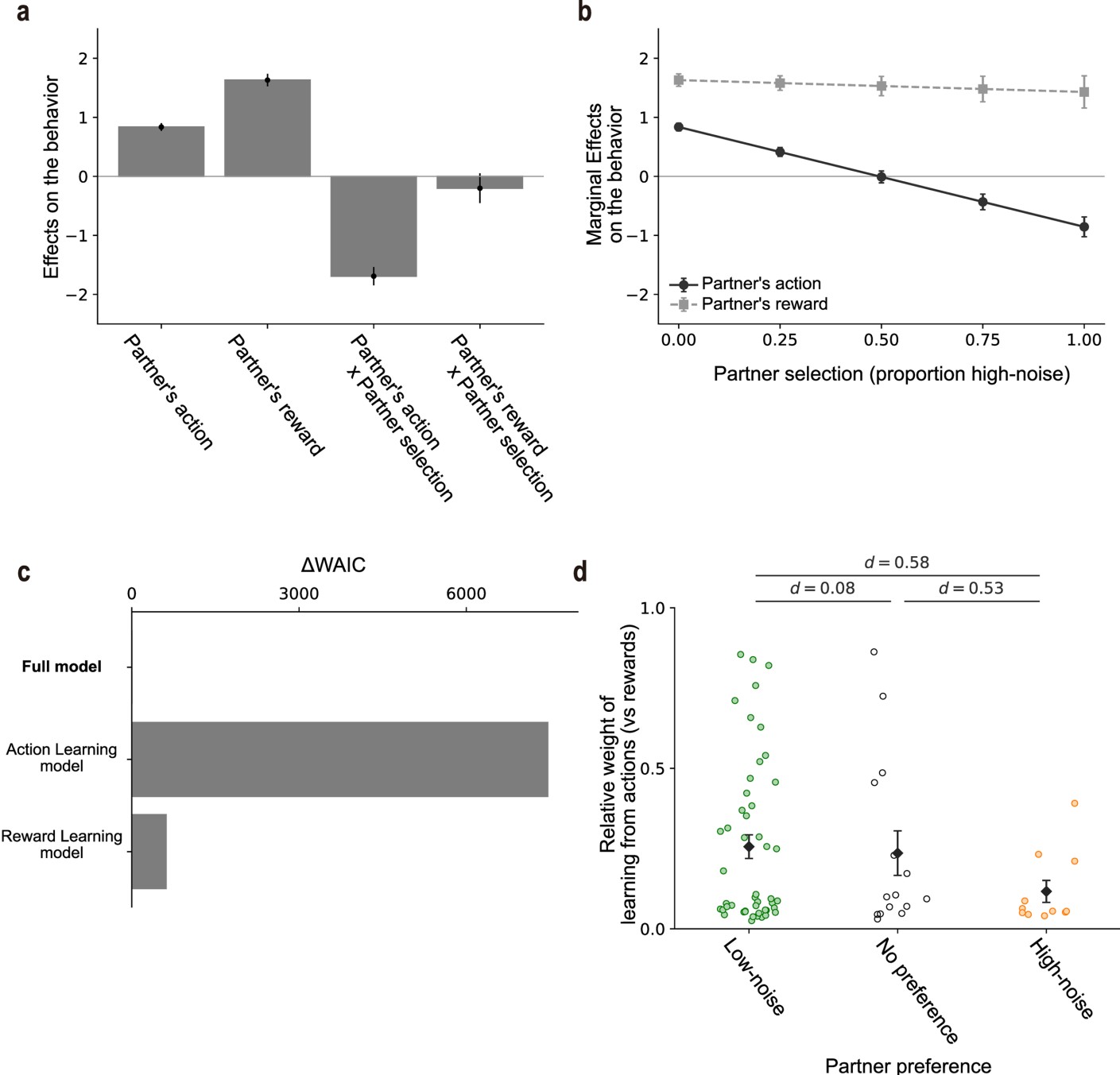

**Fig 3. Exploratory analyses of individual learning styles.** (a) Effects of partner-related variables on participants' behavior (Mean ± SEM, N = 74). The first and second bars represent the main effects of the partner's action and reward, respectively. The third bar shows the interaction between partner's action and partner selection (defined as the proportion of blocks in which the high-noise partner was chosen), indicating that imitation behavior varied with partner selection. The fourth bar reflects the interaction between partner's reward and partner selection. The means and SEMs were estimated with a generalized linear mixed effects model (GLMM). We did not report p-values as the regression analysis was exploratory. (b) Marginal effects (simple slopes) from the GLMM. Effects of the partner's action (solid) and partner's reward (dashed) evaluated at five levels of high-noise preference. Error bars show ± SEM. (c) Model comparison using Widely-applicable Akaike Information Criterion (WAIC). The Full model, which includes both action and reward learning styles, provides the best fit to the data. (d) Relative weight of learning from actions (vs rewards) estimated from the Full model, grouped by partner preference (low-noise, no-preference, high-noise). Each color dot depicts one participant (green = low-noise preference, gray = no preference, orange = high-noise preference); black diamonds and vertical lines show the group mean ± SEM. Brackets report Cohen's d for the pairwise contrasts.

of selecting the high-noise partner ($b = -1.69 \pm 0.15$). The interaction effect supports our reasoning that participants who selected the low-noise partner relied more on learning from the partner's past action compared with those who selected the high-noise partner.

This reasoning is further supported by computational modeling. We compared three computational models of observational learning based on their goodness of fit to participants' actual choice data in the Observational Learning task. We then compared parameter estimates in the best-fitted model between participants who preferred to learn from the low-noise partner and those who preferred the high-noise partner.

Inspired by previous studies on observational learning [15,16,27], our Full model assumes that an agent combines two learning systems with a certain weight: learning from the partner's actions (i.e., imitation) and learning from the partner's rewards (See Methods for details). Note that the main parameter of interest in the Full model is the weight of the two learning systems. We also tested for alternative partial models that include only one of the two learning systems (i.e., the Action Learning model and Reward Learning model).

Those models were fitted using a hierarchical modeling approach [28,29], and their goodness of fit was assessed based on the Widely-applicable Akaike Information Criterion (WAIC) [30]. The model-fitting procedure was validated through model-recovery and parameter-recovery analyses on synthesized data [31]. The model-recovery analysis confirmed the identifiability of the competing models: simulation data generated by each model were better captured by the corresponding model than by the other models, as evidenced by the confusion matrix being close to the identity matrix [31] (S2A Fig). The parameter-recovery analysis demonstrated that estimated values of the model parameters aligned well with the true generative values used in the simulation (S2B Fig). We also conducted posterior predictive checks to examine whether the Full model could reproduce the empirical behavioral patterns (i.e., the learning curve (S1C Fig) and the effects of observed reward and action on the behavior in the Observational Learning tasks (Fig 3A)), supporting that the model captures key behavioral tendencies (S3 Fig). These simulation results support the validity of our model-fitting procedure.

The model comparison indicated that the Full model provided a better fit compared to the other competing partial models (Fig 3C). We also conducted model comparisons separately for participants who preferred low-noise versus high-noise partners, and confirmed that the Full model provided the best fit in both of the groups (S4 Fig). Furthermore, we tested for a dynamic-weight model that captures flexible learning styles in which participants initially learn from observed rewards and later switch to imitation (see S1 Text for details). We found that the model did not outperform the original Full model. These findings suggest that participants employed both learning from the partner's actions and rewards for decision-making in the Observational Learning task (see S5 Fig for posterior densities of the population-level parameters), consistent with the results of the GLMM analysis (Fig 3A) and prior studies [8,9,15].

Exploring the individual differences in the parameter estimate of interest in the best-fitting Full model, we then found that the relative weight of learning from the actions (vs the rewards) was greater in participants who preferred the low-noise partner compared to those who preferred the high-noise partner (Cohen's d = 0.58; Fig 3D; see S6 Fig for comparison of other parameters). These results from the model-based analysis were consistent with our reasoning that participants who preferred the low-noise partner more, learned from the partner's actions (compared with those who preferred the high-noise partner) and that those who preferred the high-noise partner more, learned from the partner's rewards.

Finally, we examined whether and how participants adjusted their learning style according to the partner's noise level (i.e., a within-participant effect). To this end, we analyzed trial-by-trial behavior in the Observational Learning task among participants who selected both high- and low-noise partners at least once ($N = 52$). This additional GLMM analysis revealed that the reward effect was attenuated when participants observed a high-noise partner ($b = -0.113 \pm 0.025$; Fig 4). More critically, the influence of the partner's actions on participants' choices varied markedly with the partner's noise level, as reflected by a strong negative interaction ($b = -0.375 \pm 0.021$; Fig 4). This pattern indicates that the same participants relied more heavily on their partner's actions when the partner was low-noise, and suppressed imitation when the partner was high-noise.

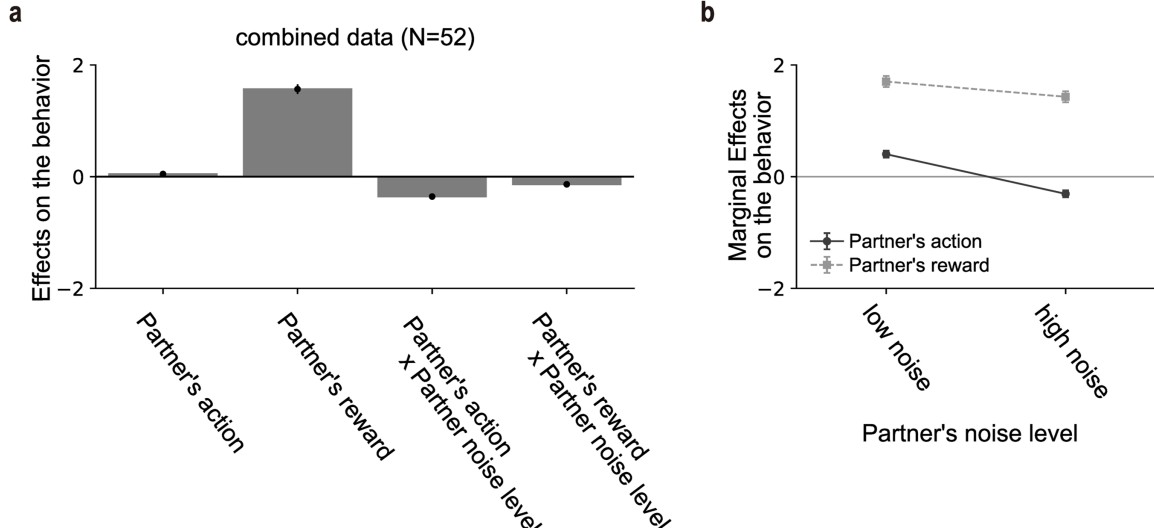

**Fig 4.** **Exploratory analyses of within-participant changes in learning style by the partner's noise level** ($N = 52$ **participants, who selected both high- and low-noise partners at least once).** (a) Effects on participants' trial-by-trial behavior in the Observational Learning task (mean $\pm$ SEM). The first and second bars represent the main effects of the partner's action and reward, respectively, while the third bar shows the interaction between the partner's action and the partner's noise level. Means and SEMs were estimated using a GLMM. P-values are not reported as the regression analysis was exploratory. (b) Marginal effects (simple slopes) from the GLMM. Effects of the partner's action (solid line) and the partner's reward (dashed line) are shown at two different levels of partner decision noise.

## Discussion

Observational learning is a fundamental cognitive ability that enables individuals to acquire knowledge and skills by observing others. Given its significance, numerous studies across various disciplines have examined the mechanisms underlying observational learning. However, few studies have directly addressed the critical question of whom individuals prefer to learn from. To address this question, we developed a novel behavioral paradigm where participants selected between two potential learning partners who differed in their level of decision noise for observational learning. We found out that participants, on average, preferred to learn from the partner with lower decision noise.

The finding that participants preferred a low-noise (high-performing) partner might initially seem intuitive and straightforward. However, from a learning and information acquisition perspective, lower decision noise, meaning less exploration, can hinder effective learning. In many learning contexts, especially those involving uncertainty, exploration is essential for identifying which actions yield the highest rewards. Without exploration, learners risk overlooking better alternatives. However, exploration also entails risks and, more importantly, can lead to missed opportunities for exploiting the option currently believed to be the best. A fundamental advantage of observational learning is that individuals can learn about the consequences of actions without being exposed to the risks associated with exploration or sacrificing exploitation opportunities. From this normative standpoint, observing a high-noise partner should be beneficial, as higher decision noise indicates greater exploration, offering more opportunities to learn about unfamiliar options without personally incurring exploration risks. Indeed, formal economic models of strategic learning support this view, demonstrating that individuals can optimize their own reward outcomes by minimizing personal exploration when observing a partner who exhibits a higher degree of exploration [19,20]. These theoretical predictions raise an important question: why did participants in our study prefer a low-noise partner?

One possible explanation is that, while exploration is necessary to solve the three-armed bandit task used in our experiment, participants may have perceived that extensive exploration was not essential for optimal performance. If the task

structure did not strongly incentivize exploration, participants may have been less motivated to select a high-noise partner. Future research could further investigate this possibility by modifying the task to increase the necessity of exploration. For instance, associations between choices and reward probabilities could be swapped midway through the task, or the reward probabilities systematically altered over time. These manipulations would potentially make a high-noise partner more valuable.

Another alternative explanation is that participants may have favored the low-noise partner because their behavior more closely matched their own decision-making style. In our data, participants' learning curves during the individual task looked very similar to those of the low-noise partner (S1A Fig vs. Fig 1E), suggesting a match in behavior. This is in line with recent research showing that learners benefit more when learning from partners whose actions are predictable [14], suggesting that perceived behavioral similarity may enhance the effectiveness of social learning.

Another interpretation is that participants may have simply chosen the partner who performed better. In our experiment, the low-noise partner earned more reward than the high-noise partner (Fig 1E), although this is not necessarily always the case, as a low-noise partner might ultimately settle on a suboptimal choice due to insufficient exploration. Prior research has shown that individuals tend to learn more quickly and effectively when learning from someone who is skilled or successful, as opposed to someone who performs poorly or inconsistently [32–35]. In our design, we manipulated the level of decision noise (i.e., random exploration), but decision noise and performance were inherently linked—lower decision noise led to better performance—making it difficult to determine whether participants' preferences were driven by the partner's exploration style or by their observed success.

To disentangle these possibilities, we estimated the effects of three potential factors on partner selection: partner performance, predictability, and information gain (proxy for directed exploration level). The results indicated that partner performance was the strongest predictor of choice, followed by predictability, while the effect of information gain had no reliable effect (Fig 2D). These findings align with prior studies that people learn more efficiently from high-performing or predictable individuals [14,32–35], suggesting that learners preferentially select partners from whom they expect to learn effectively.

To better tease apart these influences, future studies should experimentally decouple exploration from performance and predictability. One approach would be to design tasks in which partners with different levels of exploration nonetheless achieve comparable accuracy and predictability, for example, by manipulating the degree of directed exploration so that both high- and low-explorative partners obtain similar amounts of rewards. Such designs would more cleanly isolate the unique effect of exploration on partner selection.

Beyond exploration, performance, and predictability, other unmeasured factors might have affected partner selection. One illustrative factor is cognitive load: tracking a high-noise partner likely imposes greater working-memory and computational demands because participants must track a more diverse set of actions and outcomes. Consistent with this, models that include memory limits (e.g., forgetting-based Q-learning) often fit human choices better in sequential tasks [36–40], suggesting that individuals may struggle to accurately retain and update information when the number of distinct observations increases. From this perspective, a low-noise partner may be preferred simply because their behavior is easier to follow. More broadly, potential influences like cognitive load highlight the need for future work to account for a wider range of factors in partner selection.

We next examined the relation between partner selection and learning style across participants. We reasoned that those who preferred low-noise partners relied more on imitation: low-noise partners more consistently choose the currently best option, making them easier and more advantageous to imitate. To probe this possibility, we employed computational modeling analysis as well as model-free regression analysis. The results indicate that participants indeed differed in their learning styles based on partner preferences. Participants who preferred the low-noise partner had a higher weight on the Action Learning component (see Methods for details), relying more on learning from their partner's actions

rather than learning from their partner's reward outcomes, compared to those who preferred the high-noise partner (Fig 3A and 3D).

However, this analysis cannot determine whether individual differences in learning style reflect partner preferences or are induced by differences in the partner's noise level. Two possibilities remain. First, a stable tendency to imitate may drive the preference for low-noise partners. Second, the behavior of a low-noise (high-performing) partner may shift participants toward imitation more than the behavior of a high-noise (low-performing) partner.

To distinguish between these two accounts, we tested whether participants' learning style changed with the noise level of the selected partner within a participant. Restricting the analysis to participants who selected both high- and low-noise partners at least once, we estimated within-participant changes in learning style. Participants showed greater reliance on imitation when observing low-noise partners, consistent with prior studies showing selective imitation of low-noise (high-performing) partners [32–35]. This finding supports the second explanation: the behavior of a low-noise (high-performing) partner may shift participants toward imitation. However, this approach has an important limitation: because it requires exposure to both high- and low-noise partners, it includes only a subset of participants, raising the possibility of selection bias. To address this issue more directly, future work should systematically manipulate partner noise levels within participants.

An intriguing direction for future research might be exploring how social and psychological traits may impact partner preference in value-based decision making. Specifically, attributes such as perceived trustworthiness or ideological alignment could meaningfully shape preferences during observational learning. Prior research across diverse disciplines has highlighted the importance of such factors in shaping partner preference. For example, studies in mate preference have consistently demonstrated the significance of traits such as trustworthiness, attractiveness, and status in romantic partner selection [41,42]. In songbird studies, juvenile male zebra finches exposed to two adult males exhibited a preference for learning songs from tutors who displayed more aggressive behavior toward them [43]. Although not focused explicitly on preference, research in observational learning has shown that individuals are more likely to learn from others who share their political views [12]. Despite the extensive literature on partner preference in other domains, limited attention has been devoted to understanding the role of these social and relational attributes in decision-making contexts. Investigating how attributes such as trustworthiness or political alignment affect partner preference in observational learning could yield critical insights into the mechanisms of social learning and decision-making.

Another promising direction for future research involves investigating how preferences for learning partners evolve over time through repeated interactions. In the current experimental design, participants made a one-time choice between two potential partners and did not return to those options in later blocks. However, in real-world social learning scenarios, individuals often have repeated opportunities to choose whom to learn from, allowing for the gradual development of preference for a partner over time. For instance, in workplace environments, employees may initially consult multiple colleagues when tackling unfamiliar tasks. Over time, they tend to rely more on those who consistently provide accurate or helpful advice, leading to the emergence of trusted collaborators. This process reflects a dynamic updating of partner value based on repeated interactions and feedback. Investigating how such long-term relationships form and influence partner preference could provide valuable insights into the mechanisms of social learning.

In conclusion, our study investigated from whom individuals prefer to learn in observational learning contexts and found a clear preference for low-noise (high-performing) partners, contrary to theoretical predictions emphasizing the benefits of observing high-noise partners. Our computational modeling further indicates that learning style varies with partner preference: participants who preferred low-noise partners showed greater reliance on observed actions (imitation). We believe these findings offer valuable insights into the computational mechanisms underlying partner preference, contributing to our understanding of how social learning networks might be constructed and how collective behaviors emerge in social groups.

## Materials and methods

### Ethics statement

The experimental protocol was approved by the University of Melbourne Human Research Ethics Committee (Ethics ID 26997). All participants provided written informed consent before the experimental session commenced.

### Preregistration

The experimental plan was preregistered on the Open Science Framework (OSF) and can be accessed at https://osf.io/g6etf.

### Participants

Participants were recruited through advertisements on the University of Melbourne's student portal. Eligible participants were aged 18 to 35 years. Each participant received a A$10 show-up fee and additional performance-based compensation. Twenty participants took part in the pilot experiment (15 females, 4 males, and 1 unidentified; age range, 18-28 years; mean age ± SD, 22.5 ± 2.86), and 55 were recruited for the main experiment (35 females, and 20 males; age range, 18-29 years; mean age ± SD, 22.98 ± 3.25). Participants received instructions via slides and completed comprehension quizzes to ensure they correctly understood the experimental task. Every participant answered all the questions correctly.

Participants were told their partners were previous human participants in the Observational Learning tasks, but in reality, the partners were simulated agents using a Q-learning algorithm (See "Observational Learning task"). After the experiment, participants were debriefed about the deception and could withdraw their data if desired. Additionally, we conducted brief interviews to assess whether participants had suspected that the partner choices were not generated by real individuals. No participants expressed strong doubts regarding the authenticity of their partners.

We did not exclude any participants from the main dataset. For exploratory analyses, we excluded one pilot participant who completed only two of four blocks.

### Experimental design

We conducted both a pilot behavioral experiment and a preregistered confirmatory behavioral experiment, in which participants engaged in observational learning alongside partners they selected themselves (Fig 1). The experimental protocol consisted of two preparatory blocks followed by four main task blocks.

In the first preparatory block, participants independently performed a three-armed bandit task (see the "Individual Learning task" subsection below). In the second preparatory block, they participated in an observational learning task (see the "Observational Learning task" subsection below) with a pre-determined partner. During each of the main blocks, participants initially observed two potential partners executing the individual learning task (see the "Passive Observation task" subsection below). Following the observation, participants selected one of the observed individuals as their partner for the subsequent Observational Learning task (see the "Partner Selection task" subsection below). Finally, participants engaged in the Observational Learning task with their chosen partner.

**Individual learning task.** Participants performed a three-armed bandit task, where they repeatedly chose between three stimuli (fractal images) for 60 trials. Each stimulus was associated with a hidden reward probability of 0.25, 0.5, or 0.75, respectively. The associations between stimuli and reward probabilities were randomized across participants. Participants were explicitly informed that the set of stimuli, their positions on the screen, and the associated reward probabilities would remain constant within a task.

**Observational learning task.** Participants repeatedly made choices between three stimuli, not based on their reward outcomes, but by observing their partner's choices and outcomes (Fig 1C). Similar to the Individual Learning task, the three stimuli were paired with hidden reward probabilities of 0.25, 0.5, and 0.75, respectively. These associations

remained constant within a task but were randomized across blocks and across tasks. It is important to note that participants were informed that both they and their partner were faced with the same stimulus-reward associations.

In each of the 60 trials, participants first observed their partner's choice and its outcome. The chosen stimulus was highlighted with a black frame for 1 second, followed by the display of the outcome for 1.5 seconds. Afterward, a fixation cross appeared for a variable duration ranging from 1 to 2 seconds. Subsequently, participants made their own choice. The chosen stimulus was also highlighted with a black frame for 1 second. However, unlike their partner's choice, the outcome of the participant's choice was not revealed.

Crucially, in the preparatory block, participants observed a pre-determined partner. They were informed that this partner was another individual who had participated in the individual learning task on a previous day. However, the 'partner' was a simulated agent operating on a standard Q-learning algorithm with a learning rate set to 0.3 and an inverse temperature parameter of 7.0. In contrast, during the main blocks, participants selected their partners (see "Passive Observation task" and "Partner Selection task" subsections below).

**Passive observation task.** Within each block, participants completed two Passive Observation tasks. In each task, they watched a simulated partner perform 30 trials of a three-armed bandit. The two partners differed only in decision noise (undisclosed to participants): a high-noise partner simulated with a standard Q-learning model (learning rate $\alpha = 0.3$; inverse temperature $\beta = 1.5$) and a low-noise partner with inverse temperature $\beta = 20.0$ (Fig 1E). Stimulus–reward mappings and the spatial assignment of reward probabilities (0.25, 0.50, 0.75) to the three options were randomized across participants and across the two tasks within each block.

To help participants differentiate between the two potential partners, distinct real human face images were used (all of the same sex, with a multi-racial face image dataset employed to minimize potential sex and racial biases) [44]. However, to protect the identities of the individuals in the face images, we replaced the face images used in the experiment with AI-generated face drawings (created using OpenAI's ChatGPT) in Fig 1. These drawings are used solely for illustrative purposes and were not used in the actual experiments. Each participant first observed one potential partner completing 30 trials of the Individual Learning task, followed by the observation of the other partner. The presentation order of the partners was counterbalanced across the four main blocks, while their corresponding face images were counterbalanced across participants.

**Partner selection task.** For the subsequent Observational Learning task, participants chose between two potential partners: a high-noise partner and a low-noise partner, both of whom they had observed in the preceding Passive Observation task. In this task, the two facial images representing the potential partners were displayed side by side in a horizontal layout. The associations between the presentation positions of these images and the exploration levels of the potential partners were counterbalanced across blocks. Participants made their selection by clicking on the face image of the partner they preferred.

**Software.** We coded the experimental tasks using Node.js (https://nodejs.org/en) in MacOS.

## Statistical analysis

**Regression analysis.** To address whom participants preferred to learn from during observational learning, we fit mixed-effects logistic regressions with `glmer` from the `lme4` package (version 1.1-35) in R [45,46]. As preregistered, we analyzed only data from the main experiment ($N = 55$).

The dependent variable $Y$ was coded 1 if the left-hand partner was chosen in the Partner Selection task and 0 otherwise. We defined two predictors:

1. $X_1$: coded +1 if the high-noise partner appeared on the left and −1 if on the right. The coefficient $\beta_1$ indexes the effect of partner decision noise on choice (with $\beta_1 > 0$ indicating a preference for the high-noise partner).
2. $X_2$: coded +1 if the left-hand partner was presented first in the preceding Passive Observation task and −1 otherwise, controlling for presentation order bias.

The intercept captures any overall spatial bias toward choosing the left option (see Fig 1B).

We estimated two models: Model 1 included $X_1$; Model 2 additionally included $X_2$. Participants were treated as random factors, with by-participant random intercepts and random slopes for each fixed effect. We reported point estimates and standard errors of fixed effects, and assessed significance with two-tailed $t$-tests of coefficients against zero.

Using pilot data, we estimated the effect size for Model 1 (S1 Table). Based on this estimate, a bootstrap power analysis [47] indicated that a sample size of $N = 55$ provides at least 80% power to detect the Model 1 effect at a two-tailed significance level of $\alpha = 0.05$ (S2 Table).

We examined whether partner selections were better explained by factors underlying the decision noise manipulation. In the first step of this analysis, we derived three partner-level indices for each potential partner from the Passive Observation data in each block: performance, predictability, and information gain.

**Performance.** We defined performance of the partner as the total obtained reward during the Passive Observation task.

**Predictability.** The perceived predictability was defined as the extent to which the partner's observed behavior could be predicted from the participant's own decision process. To estimate the process that the participant would adopt, we used a reinforcement learning (RL) model with an information bonus parameter $\kappa \in [0, 1]$ [17], such that an information bonus term $\kappa/\sqrt{N_X}$ was added to the action value of option $X$, where $N_X$ denotes the number of times option $X$ had previously been chosen. The RL model was fit hierarchically to each participant's data from the Individual Learning task in the practice block, using the same weakly informative priors and transformation procedure as in the observational learning analysis (see Computational models subsection). Predictability was then quantified as the log likelihood that the participant-specific RL model would generate the partner's observed choices in the Passive Observation task. In other words, higher log likelihood values indicate that the partner's behavior was more consistent and predictable with the decision process estimated for that participant.

**Information gain.** The perceived information gain was defined as the extent to which the partner's observed behavior reduced the participant's uncertainty about the available options. To capture this, we used the same RL model with an information bonus parameter. In this model, the information bonus associated with an option decreases by $\kappa\left(\frac{1}{\sqrt{N_X}} - \frac{1}{\sqrt{N_X+1}}\right)$ as that option is chosen. Information gain is therefore computed as the total reduction in information bonus induced by the partner's observed behavior.

For each of the three factors, we subtracted the partner-level score (performance, predictability, or information gain) of the potential partner on the left from that of the potential partner on the right. We then replaced the partner's noise-level dummy variable $X_1$ in Model 2 with these three difference scores. All independent variables were z-standardized. In this exploratory analysis, we combined the pilot data with the main data ($N = 74$). We also reported effect sizes (standardized regression coefficients for regression analyses) and standard errors rather than p-values, as p-values are more appropriate for statistical inferences based on a priori hypotheses [25,26]. Accordingly, since all subsequent analyses were exploratory, we adopted this practice throughout.

Next, we investigated the individual learning style, that is, how the partner's actions and rewards influenced the participant's current choice within each trial. Following previous studies using a three-armed bandit task [22–24], we ran three separate GLMMs, one for each option ($X$, $Y$, and $Z$). Each GLMM estimated the probability that participants chose option $X$($Y$, or $Z$) on each trial. Each model took the following form in Wilkinson notation (using option $X$ as an example):

$$\text{logit}P(\text{choice} = X) \sim 1 + (C_t + R_t) * P + (1 + C_t + R_t | \text{ID}), \tag{1}$$

where the dependent variable was coded as 1 if the partner's choice was $X$ on trial $t$, or 0 if the participant chose $Y$ or $Z$.

Here, $C_t$ represents the partner's recent choice, coded as $+1$ if the partner chose option $X$, $-1$ if the partner chose any other option. $R_t$ captures the partner's recent outcome on trial $t$ and was coded as $+1$ if the partner was rewarded for

selecting $X$, $-1$ if rewarded for a different option, or 0 when no reward was given. The continuous variable $P$ represents the participant's preference for choosing high-noise partners, which was defined as a ratio of the number of blocks where a participant chose a high-noise partner. ID is the participant identity.

We repeated this model specification for option $Y$ and option $Z$. We derived a mean effect of each predictor across the three models. Specifically, we obtained the variance-weighted average of its coefficients (and corresponding standard errors) across $X$, $Y$, and $Z$ models and its variance.

To test whether learning style varied with the noise level of the currently chosen partner within a participant, we fit a within-participant GLMM analogous to the option-wise models above. For participants who chose both high- and low-noise partners at least once, we ran three separate mixed-effects logistic regressions (one per option $X$, $Y$, $Z$) and then pooled coefficients across options by variance-weighted averaging. Using option $X$ as an example, the model in Wilkinson notation was:

$$\text{logit}\, P(\text{choice} = X) \sim 1 + (C_t + R_t) * D + (1 + (C_t + R_t) * D \mid \text{ID}), \qquad (2)$$

where $C_t$ indicates whether the partner chose $X$ on trial $t$ ($+1$ if the partner chose $X$, $-1$ otherwise), $R_t$ encodes the partner's reward on trial $t$ relative to $X$ (as defined above), and $D$ is a block-level indicator of the currently chosen partner's decision-noise level (effect coded: $D = +1$ for high-noise, $D = -1$ for low-noise). Random intercepts and random slopes for $C_t$ and $R_t$ were included for participant (ID).

**Computational models.** To identify a participant's learning style for observational learning, we considered three computational models. These models were fitted to the choice data in the Observational Learning task and compared based on their goodness of fit. We then examined differences in parameter estimates from the best-fitting model between participants who chose to learn from the low-noise partner and those who opted for the high-noise partner.

**Full model.** This model integrated two learning styles: learning from a partner's actions and learning from a partner's rewards. Specifically, it learned the partner's action tendencies (i.e., which action the partner tends to choose) from their actions and learned action values of available options from their reward outcomes through reinforcement learning. The model assigns two key quantities to each option $X, Y, Z$:

- **Action values** ($V_X, V_Y, V_Z$) representing learned action values from observational learning.
- **Action tendencies** ($A_X, A_Y, A_Z$) capturing the tendency to imitate a partner's actions.

Each of these is initialized as follows: $V_X = V_Y = V_Z = 1/2$ and $A_X = A_Y = A_Z = 1/3$.

The model updates these values separately, following the learning rules of its two components: Reward Learning and Action Learning components. The Reward Learning component updates action values based on observed rewards. If a partner chooses option $X$ on trial $t$, the update follows:

$$V_{X,t} \leftarrow V_{X,t-1} + \alpha_V(R_t - V_{X,t-1}), \qquad (3)$$

where $R_t \in \{0, 1\}$ is the observed reward, and $\alpha_V \in [0, 1]$ is the learning rate.

The Action Learning component updates action tendencies based on a partner's choices, reinforcing chosen actions while decreasing the tendency for unchosen ones:

$$A_{\text{chosen},t} \leftarrow A_{\text{chosen},t-1}$$
$$+ \alpha_A(1 - A_{\text{chosen},t-1}), \qquad (4)$$
$$A_{\text{unchosen},t} \leftarrow A_{\text{unchosen},t-1}$$
$$+ \alpha_A(0 - A_{\text{unchosen},t-1}). \qquad (5)$$

Here, $\alpha_A \in [0, 1]$ is the learning rate for action tendencies.

To determine which option to choose, the model combines both learned action values and action tendencies using a weighting parameter $w_A \in [0, 1]$:

$$Q_X = (1 - w_A) \cdot V_X + w_A \cdot A_X. \qquad (6)$$

The probabilities of selecting an option were calculated using the softmax rule:

$$\Pr(\text{choice} = X) = \frac{\exp(\beta \cdot Q_X)}{\sum_j \exp(\beta \cdot Q_j)} \qquad (7)$$

where $\beta \geq 0$ is an inverse temperature parameter controlling decision-noise (higher $\beta$ leads to more deterministic choices).

**Action learning model.** This model only learned from a partner's actions to imitate their choices. It is a partial model in a way that it lacked the Reward Learning component of the Full model. The action tendency updates were done in the same manner as the Full model. The choice probabilities were computed in the same manner except this model used the action tendency $A$ instead of the combined value $Q$.

**Reward learning model.** This model only learned from a partner's rewards. It is also a partial model because it lacked the Action Learning component compared to Full model. The action value updates occurred in the same manner as Full model. The choice probabilities were calculated using the softmax rule except this model only used the action value $V$ instead of $Q$.

**Model fitting.** To fit the different computational models to participants' choice data in the Observational Learning task, we employed a Bayesian hierarchical modeling approach. This was done using Markov chain Monte Carlo (MCMC) sampling implemented in Stan (version 2.35.0) via the Python interface CmdStanPy (version 1.2.1). Four independent chains were run with 1,000 warm-up iterations followed by 1,000 sampling iterations, resulting in a total of 4,000 posterior samples per parameter.

In this hierarchical framework, individual-level parameters are first defined on an unconstrained (real-valued) scale and then transformed to respect their respective domains. For parameters constrained to the unit interval, such as the learning rate for imitation $\alpha_A$, we define raw parameters $\alpha_{A,\text{raw}}$ on the real-valued line and apply a logistic transformation:

$$\alpha_A = \frac{1}{1 + e^{-\alpha_{A,\text{raw}}}}, \qquad (8)$$

to ensure that $\alpha_A \in [0, 1]$. For strictly positive parameters, such as the inverse temperature $\beta$, we define a raw parameter $\beta_{\text{raw}}$ on the real-value line and map it via an exponential transformation:

$$\beta = \exp(\beta_{\text{raw}}). \qquad (9)$$

At the group level, the raw parameters are modeled as drawn from normal distributions:

$$\theta_{\text{raw},i} \sim \mathcal{N}(\mu_\theta, \sigma_\theta^2), \qquad (10)$$

where $\theta$ stands in for any transformed parameter (e.g., $\alpha_A$, $\alpha_V$, or $\beta$). Hyperparameters are given weakly informative priors:

$$\mu_\theta \sim \mathcal{N}(0, 1), \qquad (11)$$

$$\sigma_\theta \sim \text{Half-Cauchy}(0, 3), \qquad (12)$$

a choice that enforces non-negativity while allowing for substantial variation in individual-level estimates [28].

**Model recovery analysis.** To ensure the robustness and reliability of our model comparison, we performed a model recovery analysis [31]. Simulated choice data were generated using each model under experimental conditions mirroring the study design (74 participants, 60 trials per block, and four blocks). The simulated data assumed that the probability of a partner being low explorative was 0.7, and the probability of a partner being high explorative was 0.3, reflecting the empirical partner selection distribution (Fig 2A and 2B).Parameter values were drawn from Beta distributions (Beta(1.1, 1.1)) for learning rates and the weight and a Uniform distribution ($U(0, 30)$) for the inverse temperature.

Each simulated dataset was then fit using all three candidate models, and WAIC was calculated to identify the best-fitting model. This procedure was repeated 40 times, allowing us to construct a confusion matrix representing the proportion of times the generating model was correctly identified (S2A Fig).

**Parameter recovery analysis.** To further assess the validity of our parameter estimates, we conducted a parameter recovery analysis using the Full model, which was identified as the best-fitting model. Simulated choice data were generated using parameter values drawn from the same population-level distributions described above. The Full model was then fit to the simulated data using the same MCMC procedure.

We were interested in individual weighting parameters for imitation $w_A$ (see Computational models), so we evaluated individual parameter recovery by computing Pearson correlation coefficients between the true simulated parameters and the corresponding estimated parameters. This process was repeated 40 times to ensure robust results. Strong positive correlations would indicate reliable parameter recovery, supporting the validity of the model's parameter estimates (S2B Fig).

To test whether differences in partner noise-level preference could affect parameter recoverability of the Full model, we also conducted a group-stratified assessment. Simulated participants were divided into high-noise preference (choosing high-noise partners more than twice) and low-noise preference (choosing low-noise partners more than twice) groups, discarding no-preference participants (choosing high- and low-noise partners equally). We computed the same parameter-wise Pearson correlations between true and estimated values, and additionally examined the distribution of signed estimation errors separately within each group (S2C Fig). Strong positive correlations in both groups, coupled with estimation error distributions centered near zero, indicate that parameter recovery works equally well across the two noise-level preference groups.

**Posterior predictive checks.** We conducted posterior predictive checks to assess whether the Full model could reproduce key behavioral patterns. Specifically, we examined whether model-generated choices recovered the learning curve (S1C Fig) and the effects of the partner's actions and rewards on the participant's behavior in the Observational Learning task (Fig 3A). To this end, we drew individual parameter estimates for all 74 participants from the fitted Full model and simulated choice data for four blocks of the task under the same empirical conditions (trial structure, reward contingencies, and partner noise level). We then computed the learning curve averaged across participants (S3A Fig). In addition, we fitted the simulated choice data using the same GLMM used for the empirical data analysis (Fig 3A). Repeating this procedure 500 times yielded the means and standard deviations of the fixed-effect estimates (S3B Fig).

## Supporting information

**S1 Text. Dynamic-weight model.**
(DOCX)

**S2 Text. Correlation between participants' noise level and partner preference.**
(DOCX)

**S1 Table. Partner selection GLMM results: pilot and main experiments.** Reported are fixed-effect estimates, SEM, t-statistics, and p-values. Negative coefficients indicate a decreased likelihood of selecting the higher-noise partner. (XLSX)

**S2 Table. Power analysis for the partner selection GLMM.** Power was computed based on the estimated effect size in Model 1, which included only the partner's noise level as a predictor (see S1 Table). Statistical power estimates are reported for different sample sizes at a two-tailed significance threshold of $\alpha = 0.05$. (XLSX)

**S1 Fig Proportions of correct choices.** Each panel contains two plots: on the left, the proportion of correct choices across trials (chance level = 1/3); on the right, overall accuracy (mean ± SEM). (a) Individual Learning (main study, practice block). Accuracy increased over trials. (b) Observational Learning (main study, practice block). Accuracy increased over trials. (c) Observational Learning with selected partner (main study; $N = 55$). Accuracy (mean across 4 blocks ± SEM) increased over trials. (d) Observational Learning (main + pilot), stratified by partner preference. High-noise (orange dashed) vs. low-noise (green solid). (e) Observational Learning (main + pilot), stratified by block. Results for Blocks 1–4 (distinct colors). (TIFF)

**S2 Fig Model and parameter recovery analyses.** (a) Confusion matrix showing the proportion of times each generative model (rows) was correctly identified by model fitting (columns) based on WAIC. Simulated datasets were generated under conditions matching the empirical design (74 participants, 60 trials per block, 4 blocks) and repeated 40 times. (b) Parameter recovery for the full model. Top: Scatter plots show the correspondence between true (simulated) and estimated values for four parameters: the relative weight of learning from the partner's action (vs reward), the learning rate for the partner's action, the learning rate for the partner's reward, and the inverse temperature (log scale). Bottom: Histograms of Pearson correlation coefficients between true and estimated parameters across 40 simulations. (c) Distribution of parameter-recovery correlations (left) and biases (right) stratified by partner preference. Each row corresponds to one recovered parameter: the relative weight of learning from the partner's action (vs reward), the learning rate for the partner's action, the learning rate for the partner's reward, and the inverse temperature (log scale). For correlations (two left columns), each histogram shows the distribution of Pearson correlation coefficients between true and estimated parameters across 40 simulated datasets, separately for high-noise preference and low-noise preference groups. For biases (two right columns), each histogram shows the distribution of signed estimation errors (estimated - true) under the same stratification. Correlation distributions are strongly shifted toward positive values in both groups, and bias distributions are tightly centered around zero, indicating reliable recovery without systematic over- or under-estimation in either stratum. (TIFF)

**S3 Fig Posterior predictive checks for the Full model.** (a) Simulated learning curve and overall accuracy in the Observational Learning task (S1C Fig). Left panel: the simulated proportion of correct choices (mean across 4 blocks ± SEMs). Right panel: simulated overall accuracy (Mean ± SEM) (b) Simulated effects of partner-related variables on participants' behavior (Mean ± SEM). For different simulated choice data, we re-fit the same generalized linear mixed-effects model (Fig 3A) 500 times, obtaining the means and SEMs of simulated effects. (TIFF)

**S4 Fig Model comparison stratified by partner preference.** We conducted model comparisons separately for participants who preferred low-noise partners and those who preferred high-noise partners (Fig 2A and 2B in the main text). The Full model provided the best fit according to WAIC in both of the groups. (TIFF)

**S5 Fig Posterior population-level distributions of parameters in the Full model.** Density plots show the population-level posterior distributions for each parameter estimated in the Full model: the relative weight of learning from the partner's action (vs reward), the learning rates for the partner's action and reward, and the inverse temperature (log scale). (TIFF)

**S6 Fig Individual parameter estimates from the Full model, stratified by partner preference (Low-noise, No preference, High-noise).** Left: the learning rate for the partner's action (imitation); middle: the learning rate for the partner's reward; right: the inverse temperature (log scale). Dots show participants; black diamonds indicate group means ± SEM. Brackets report Cohen's d for pairwise contrasts. (TIFF)

## Acknowledgments

We thank Michele Garagnani for helpful comments, Elizabeth Bowman for facilitating the experiments, and our colleagues in the lab at the Centre for Brain, Mind and Markets for their support.

## Author contributions

**Conceptualization:** Gota Morishita, Shinsuke Suzuki.

**Data curation:** Gota Morishita, Shinsuke Suzuki.

**Formal analysis:** Gota Morishita, Shinsuke Suzuki.

**Funding acquisition:** Carsten Murawski.

**Investigation:** Gota Morishita.

**Methodology:** Gota Morishita, Carsten Murawski, Shinsuke Suzuki.

**Project administration:** Gota Morishita, Carsten Murawski, Shinsuke Suzuki.

**Resources:** Gota Morishita, Carsten Murawski, Shinsuke Suzuki.

**Software:** Gota Morishita, Shinsuke Suzuki.

**Supervision:** Carsten Murawski, Nitin Yadav, Shinsuke Suzuki.

**Validation:** Gota Morishita, Carsten Murawski, Nitin Yadav, Shinsuke Suzuki.

**Visualization:** Gota Morishita, Shinsuke Suzuki.

**Writing – original draft:** Gota Morishita, Shinsuke Suzuki.

**Writing – review & editing:** Gota Morishita, Carsten Murawski, Nitin Yadav, Shinsuke Suzuki.

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
