## [Decision Letter · Decision Letter 0]

15 Jul 2025

PCOMPBIOL-D-25-00945

Whom Do We Prefer to Learn From in Observational Reinforcement Learning?

PLOS Computational Biology

Dear Dr. Morishita,

Thank you for submitting your manuscript to PLOS Computational Biology. After careful consideration, we feel that it has merit but does not fully meet PLOS Computational Biology's publication criteria as it currently stands. Therefore, we invite you to submit a revised version of the manuscript that addresses the points raised during the review process.

Please submit your revised manuscript within 60 days Sep 14 2025 11:59PM. If you will need more time than this to complete your revisions, please reply to this message or contact the journal office at ploscompbiol@plos.org. Please include the following items when submitting your revised manuscript:

We look forward to receiving your revised manuscript.

Kind regards,

Bastien Blain

Academic Editor

PLOS Computational Biology

Natalia Komarova

Section Editor

PLOS Computational Biology

**Journal Requirements:**

At this stage, the following Authors/Authors require contributions: Gota Morishita. Please ensure that the full contributions of each author are acknowledged in the "Add/Edit/Remove Authors" section of our submission form.

4) We notice that your supplementary Figures are included in the manuscript file. Please remove them and upload them with the file type 'Supporting Information'. Please ensure that each Supporting Information file has a legend listed in the manuscript after the references list.

5) 1a, 1b, and 1c include image of an identifiable person. Please provide written confirmation or release forms, signed by the subject(s) (or their guardian), giving permission to be photographed and to have their images published under a Creative Commons license. You may upload permission forms to your submission file inventory as item type 'Other'. Otherwise, we kindly request that you remove the photograph.

Potential Copyright Issues:

i) Please confirm (a) that you are the photographer of 1a, 1b, and 1c, or (b) provide written permission from the photographer to publish the photo(s) under our CC BY 4.0 license.

**Reviewers' comments:**

Reviewer's Responses to Questions

Reviewer #1: Employing behavioral experiments and computational modelin, this study investigates how people choose whom to learn from in observational learning. The authors tested this question using a pilot and a preregistered experiment and found that people prefer to observe low-noise rather than high-noise individuals. Further analysis suggested that individuals who preferred low-noise models tended to use imitative learning, whereas those who preferred high-noise individuals tended to learn from others’ reward outcomes.

The research question is interesting and deserves attention in the field. Although the methods and analyses were conducted rigorously, there are several concerns that need to be addressed:

1. The most important finding in this study is the preference for low-noise demonstrators in observational learning. However, low- and high-noise demonstrators are not clearly distinguished from high- and low-performing demonstrators. Regardless of noise level, choosing better-performing demonstrators is an optimal strategy. Although the authors clearly address this issue in the Discussion, both the title and abstract may mislead readers into interpreting the findings as solely about noise preference rather than performance-based preference.

2. The conclusion that individuals who preferred low-noise demonstrators relied on imitation may require further analysis for confirmation. In a stable environment, once participants identify a demonstrator’s performance level, the need for ongoing evaluation may diminish, as they have already determined who is worth learning from. Additionally, the actions of better-performing demonstrators are more consistently tied to better outcomes in later trials, in contrast to the actions of high-noise (or worse-performing) demonstrators. It is therefore possible that participants initially learn from observed outcomes and later switch to imitation, a dynamic not fully captured in the current analyses. This possibility should at least be discussed.

3. Does Figure S4 indicate that higher noise levels of the participants themselves in the individual learning task correlate positively (and significantly) with preference for the high-noise demonstrator?

Minor points

Legend to Figure 1d, last sentence: Please add «of blocks» after «The order»

Please check figure numbering in the text throughout (there are for example multiple references to Figure 2 that should refer to Figure 1).

Reviewer #2: In this manuscript, the authors investigate two interesting aspects of observational learning (OL): (i) whether people prefer to learn from a high-noise or a low-noise partner after passively observing their performance on a three-armed bandit task, and (ii) whether individual preferences in partner preference relate to observational learning strategy (imitation vs vicarious reward learning) in a subsequent active OL task. They find that overall, participants exhibit a preference for low-noise partners, and that this preference is associated with more imitative learning.

These results are interesting and novel, given that partner selection during observational learning has not, to my knowledge, been investigated. I have however, several concerns about the claims related to exploration and am generally unsure if the study design is appropriate for testing the questions of interest.

Major concerns:

1) My main concern, which is raised by the authors during the discussion, but not before, is that there are multiple confounds associated with the noise manipulation (i.e. participants observing and choosing between a low-noise partner and a high-noise partner): cognitive effort (more effort needs to be deployed to learn from a high-noise partner), reward obtained being another one (the low-noise partner performs better), and/or choice reliability/predictability. This makes it virtually impossible to know what factor actually drives partner preference; yet, until the discussion the authors claim that what they manipulate with this is the partner’s exploratory tendencies. I think such claim is misleading, and the authors should find a way to at least distinguish exploration from the other confounds (either with additional analyses if possible, or a better designed paradigm). Otherwise, the interpretability of the findings is too low.

In other words, I do believe that there is great premise in operationalizing the preference for low noise vs high exploration (since many real life situations involve this kind of “dilemma”), but I don’t think that the current paradigm and analyses tackle this well given that high noise doesn’t necessarily mean high exploration. The type of “adventurous” exploratory partner the authors describe is likely rarely random in their exploration; therefore, it would seem key to operationalize an exploratory partner that isn’t just driven by high noise, but with directed exploration aiming to maximize information gain.

A potential additional analysis (though I am happy for the authors to find other ways to address the concern) could be to compute behavioral indices of rewards obtained, information gain, action predictability and/or cognitive effort during the passive observation task (maybe extracting those variables from an optimal learner agent since no data is available from the participants during that phase), and use these variables as predictors of partner selection in the first GLMM. This would not only help disentangle potential drivers of partner preference, but also provide some insights into the computational mechanisms involved during passive observation.

2) The authors’ initial hypothesis about the individual differences “when individuals primarily rely on imitation, they may show a preference for low noise partners”. Yet, individuals’ OL style is not assessed before they are asked to choose a partner. It would be helpful to further justify why the study was designed that way (i.e. without a possibility to assess participants’ learning style at baseline). While this study is not designed to test causality, the hypothesis above is qualitatively different from the finding that “individuals who preferred low-noise partners demonstrated a greater reliance on imitative learning”.

3) More generally, it would be interesting to further discuss whether both learning style and partner preferences are likely to vary depending on the context (and thus, it’s likely because participants chose a low-noise partner that they rely more on imitation), or whether they are more stable and trait-like. Maybe some exploratory behavioral analyses on participants who show some variability in their partner selection across blocks could speak to that? For example, do the same participants exhibit higher partner’s action effect/lower partner’s reward effect in the blocks where they chose the low-noise partner compared to blocks where they chose the high-noise partner? If so it might be interesting to test a computational model in which the weight parameter varies depending on the chosen partner.

4) Given that the 0.25, 0.5, 0.75 probabilistic structure was the same across both passive and active tasks, is it possible that participants could have learned by mapping that structure across the two tasks (even with the stimuli being different)? Was the position of the 0.25, 0.5 and 0.75 stimuli different between the blocks? It would be helpful to mention this probabilistic structure in Figure 1 and/or in the overview of the experimental design, since this is important information.

5) In addition to picking the best model and comparing the parameter values, it would further strengthen the claim if the authors could show that participants who prefer the low-noise partner (from Figure 2a-b) are best fit by the action learning model, while participants who prefer the high-noise partner are best fit by the reward learning model.

6) While it was great to see robust parameter and model recovery, it would also be great if the authors could perform posterior predictive checks to ensure the model captures behavioral tendencies. This could include reproducing the learning curves from model-generated data, and reproducing the GLMM effects from Figure 3a.

7) I assume it is more difficult for the model to separate the two learning strategies when it’s learning from the low-noise partner (because the action and reward are confounded on a higher proportion of trial). Is this true? If so, could that be impacting the individual difference results in any way?

Minor comments:

8) It would be helpful if the legend of Figure 1 could mention the number of trials in each task.

9) Was the presentation order of the two potential partners during the passive observation task also counterbalanced across blocks within participants? According to the methods, it seems like this was counterbalanced across blocks but this wasn’t made clear when first describing the task.

10) Basic behavior: please also report the proportion and correct choices and learning curves for the main OL task (Figure S1c) broken down by blocks and/ or by selected partner for completeness.

11) Figure 3. For illustrative purposes, it would be helpful to extract and plot the marginal means of the two interaction effects from the GLMM in panel a. This would allow the reader to visualize the effects of partner’s action and partner’s reward separately for each of the 5 partner selection levels.

12) Figure 3c. Is this effect specific to the action learning weight? Is it possible that participants in the low-noise partner selection group are simply better learners and therefore also show better performance and better reward learning? I understand from Figure S4 that other parameters do not show the same group differences; however, it would be helpful to also compare behavioral metrics such as overall performance or learning curves.

13) Since there is only one weight parameter in the model it would be helpful to rename Figure 3c y-axis with something that reflects that this is not just the weight of learning from the partner’s action but the relative weight of learning from actions versus rewards. Same comment for Fig S2b title of left panel.

14) Model/parameter recovery (Figure S2) why was the number of simulations different for the two analyses (30 vs 40 simulations). The y-axis title of Fig S2b bottom row suggests that parameter recovery was also 30 simulations, which is inconsistent with the legend and methods.

13) Figure S2a. Please write the exact numbers on the matrix as it is difficult to match them to the color scale. Especially the action learning model seems less well recovered than the other two but it is unclear what it might be confused with.

14) For the plots in Figure 3c and Figure S4, it would be helpful to depict the mean value of the parameter (or mean of the posterior) so it’s easier to visualize the direction of the differences between groups.

15) Typo: Figure 3 legend – “explanatory” should be “exploratory”.

16) The preregistration mentions some exclusion criteria but those are not reported in the Methods. Did any participant meet any of the exclusion criteria?

17) While I appreciate the preregistration effort, I believe the authors should downplay the extent to which the experiment and analyses were preregistered. This is for a few reasons: the preregistration does not report the findings of the pilot study, the observational learning task that was administered after the partner selection task is not described (while I understand the analyses pertaining to this task were exploratory, it seems strange not to have preregistered the task itself and overall protocol prior to data collection), the power analysis is based on equation (1) for the GLMM which is not the analysis that was conducted in the current manuscript (only one predictor of partner selection compared the three), and the power analysis is missing the effect size (which was presumably extracted from the pilot study, but that is not clear). I think it’s acceptable to deviate from a preregistration, but this should be more clearly explained.

**Have the authors made all data and (if applicable) computational code underlying the findings in their manuscript fully available?**

Reviewer #1: None

Reviewer #2: Yes

PLOS authors have the option to publish the peer review history of their article (what does this mean?). If published, this will include your full peer review and any attached files.

Reviewer #1: No

Reviewer #2: **Yes: **Caroline J Charpentier

**Figure resubmission:**
---

## [Decision Letter · Decision Letter 1]

19 Nov 2025

PCOMPBIOL-D-25-00945R1

Whom Do We Prefer to Learn From in Observational Reinforcement Learning?

PLOS Computational Biology

Dear Dr. Morishita,

Thank you for submitting your manuscript to PLOS Computational Biology. After careful consideration, we feel that it has merit but does not fully meet PLOS Computational Biology's publication criteria as it currently stands. Therefore, we invite you to submit a revised version of the manuscript that addresses the points raised during the review process.

We look forward to receiving your revised manuscript.

Kind regards,

Bastien Blain

Academic Editor

PLOS Computational Biology

Natalia Komarova

Section Editor

PLOS Computational Biology

**Journal Requirements:**

We ask that a manuscript source file is provided at Revision. Please upload your manuscript file as a .doc, .docx, .rtf or .tex. If you are providing a .tex file, please upload it under the item type u2018LaTeX Source Fileu2019 and leave your .pdf version as the item type u2018Manuscriptu2019.

**Reviewers' comments:**

Reviewer's Responses to Questions

**Comments to the Authors:**

Reviewer #1: Thank you for providing a response revision. Regarding previous major point 3, it would be helpful to also specify the p-value of the correlation, even if this more explorative.

Reviewer #2: The authors have addressed all my concerns, and I am happy to recommend this manuscript for publication.

**Have the authors made all data and (if applicable) computational code underlying the findings in their manuscript fully available?**

Reviewer #1: None

Reviewer #2: Yes

PLOS authors have the option to publish the peer review history of their article (what does this mean?). If published, this will include your full peer review and any attached files.

Reviewer #1: No

Reviewer #2: **Yes: **Caroline J. Charpentier

**Figure resubmission:**
---

## [Editor Report · Decision Letter 2]

28 Nov 2025

Dear Mr. Morishita,

We are pleased to inform you that your manuscript 'Whom Do We Prefer to Learn From in Observational Reinforcement Learning?' has been provisionally accepted for publication in PLOS Computational Biology.

Best regards,

Bastien Blain

Academic Editor

PLOS Computational Biology

Natalia Komarova

Section Editor

PLOS Computational Biology

---

## [Editor Report · Acceptance letter]

PCOMPBIOL-D-25-00945R2

Whom Do We Prefer to Learn From in Observational Reinforcement Learning?

Dear Dr Morishita,

I am pleased to inform you that your manuscript has been formally accepted for publication in PLOS Computational Biology. Your manuscript is now with our production department and you will be notified of the publication date in due course.

With kind regards,

Anita Estes
